# A conformational change in α-catenin's actin-binding domain governs adherens junction maturation
Lukas Windgasse ⓘ & Carsten Grashoff ⓘ ✉

The formation and maintenance of epithelia is critical for animal development and survival. Central to epithelial integrity are cadherin-based complexes called adherens junctions (AJs), which form physically robust but inherently dynamic cell-cell adhesions. How AJs function at the molecular level remains incompletely understood because techniques to study the central AJ proteins within the dynamic adhesion structure are scarce. Using a conformation sensitive probe that is amenable to fluorescence lifetime and anisotropy imaging, we demonstrate that the maturation of AJs is accompanied by a conformational change in the actin-binding domain of α-catenin. The structural transition depends on the degree of junctional maturation and requires actin polymerisation, but it is insensitive to vinculin binding to α-catenin. These different conformational states correlate with distinct α-catenin mobilities, with α-catenin unexpectedly showing an overall increased protein turnover in mature AJs. Collectively, the data reveal that α-catenin undergoes a previously proposed C-terminal conformational transition during epidermal differentiation to form mechanically stable yet dynamic cell-cell adhesions.

The mechanical integrity of epithelial tissues depends on the formation and maintenance of physically robust cell–cell adhesions that can withstand both internally generated and externally applied stresses. Conversely, intercellular junctions must be dynamic to enable efficient tissue responses, for example, during development and homeostasis, but also during wound healing after tissue injury[1,2]. It has been established that these properties are largely mediated by cell adhesion complexes, known as adherens junctions (AJs), which physically connect neighboring cells via cadherin receptors and simultaneously link to the intracellular actin cytoskeleton via β- and α-catenin[3]. However, a molecular understanding of how AJs work, in particular how they absorb mechanical stresses while maintaining their dynamic properties, has remained elusive due to a lack of suitable techniques to study the individual AJ components within the adhesion structure in living cells.

Central to the function of these type of cell–cell adhesion appears to be the AJ-resident protein α-catenin, which comprises an N-terminal β-catenin-binding domain (ND), a central modulation domain (MD) containing a force-sensitive vinculin-binding site, and a C-terminal actin-binding domain (ABD; Fig. 1a). Previous reports have proposed that force-dependent conformational changes in the MD and a subsequent recruitment of vinculin are central to α-catenin's role as an integrator of mechanical signals at cell–cell junctions[4–7], and a biosensor has been developed to monitor this conformational switch in living cells[8]. Intriguingly, more recent studies also point to a specific function of two α-helices (H0 and H1) in the C-terminal ABD (Fig. 1a), which alter their conformation upon actin engagement[9]. Consistent with an earlier study showing that the cadherin-catenin complex binds actin filaments under mechanical loads[10], force spectroscopy experiments showed the critical role of the α-catenin ABD, and in particular the H0 and H1 helices, in mediating this catch-bond-like behavior[11]. Subsequent studies demonstrated that α-catenin–actin catch-bonding depends on the orientation of the mechanical forces applied through actin and that is a result of the cooperative action of multiple molecules[12,13]. Collectively, these studies suggest that the complex of cadherin, α/β-catenin and f-actin is allosterically stabilized under mechanical load, and that this stabilization effect depends on a structural rearrangement at the C-terminus of α-catenin. Whether, and under what conditions, the structural transition in the H0 and H1 domains occurs in cells remains incompletely understood.

The experiments presented below demonstrate that the putative conformational change in the ABD of α-catenin does occur in cells and correlates with AJ maturation during epithelial differentiation. By combining a Förster resonance energy transfer (FRET)-based biosensor with fluorescence lifetime imaging microscopy (FLIM) and fluorescence anisotropy measurements, we show that, in contrast to the previously described structural rearrangement of the MD domain, the H0/H1 switch in the ABD is insensitive to direct vinculin binding but requires actin polymerization. Unexpectedly, our data show that the actin-bound, presumably stabilized α-catenin catch-bond

University of Münster, Institute of Integrative Cell Biology and Physiology, Münster, Germany. ✉e-mail: grashoff@uni-muenster.de

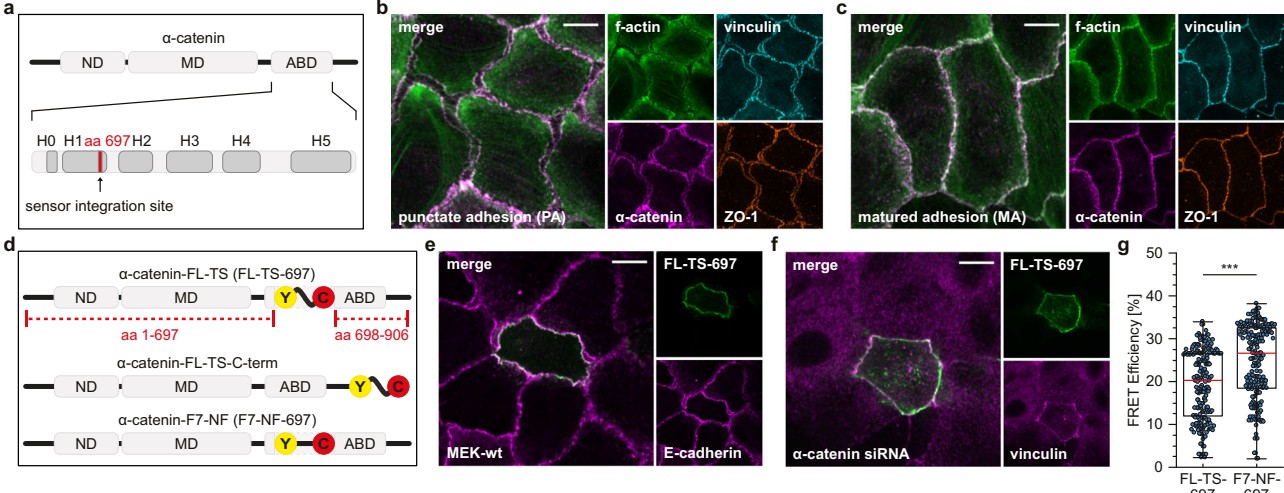

**Fig. 1 | Investigating α-catenin during the formation and maturation of AJs.**
**a** Simplified domain structure of α-catenin showing the N-terminal domain (ND), the modulation domain (MD), and the actin-binding domain (ABD). The ABD comprises six α-helices; H0 and H1 modulate actin-binding affinity.
**b**, **c** Representative images of differentiating MEKs fixed and immunostained at early (**b**) and late stages (**c**) of AJ formation; merged images display signals from the f-actin and α-catenin channels. Note the heterogeneous f-actin network in punctate adhesions (PA), which reorganizes towards a cortical localization in matured adhesions (MA). **d** Force-sensitive (FL-TS) and force-insensitive (F7-NF) FRET modules were inserted after aa 697. As an additional control, the FRET module was fused to the C-terminus of α-catenin (FL-TS-C-term). **e** Representative image of a differentiated MEK expressing the FL-TS-based α-catenin sensor (green) co-stained for E-cadherin (magenta). **f** Representative image of an α-catenin-depleted MEK expressing the FL-TS-based α-catenin sensor (green). Note the rescue of vinculin (magenta) recruitment in α-catenin sensor-expressing cells. **g** Live-cell FLIM data of differentiated MEKs expressing either FL-TS- or F7-NF-based α-catenin constructs showing a wide range of FRET efficiencies in a mixed population of cells ($N = 6$ replicates, $n = 206, 209$ cells). Boxplots show the median, the 25th and 75th percentiles and whiskers reaching to the last data point within 1.5× interquartile range. Scale bars: 10 μm. Two-sample KS test: ***$p < 0.001$.

conformation correlates with an overall increased rather than decreased protein turnover at cell–cell junctions. Together with the previous models, our data suggest that the structural adaptation of α-catenin's C-terminal ABD may be key to the dynamic but mechanically resilient properties of AJs. The genetically encoded biosensor established here should be highly useful for investigating this effect in a variety of model systems.

## Results

The epidermis is a multilayered stratified epithelium that undergoes morphologically distinguishable transitions during development that can be mimicked in cell culture experiments using basal keratinocytes[14]. In the early stages of cell–cell adhesion formation, AJs are characterized by a punctate, zipper-like morphology with heterogeneous α-catenin intensities, partially overlapping with cross-cellular f-actin fibers (Fig. 1b). Mature AJs typically show more homogeneous α-catenin signals at intercellular contacts, with a reorganized actin cytoskeleton exhibiting a prominent cortical localization (Fig. 1c). Prolonged cell culture eventually leads to epithelial stratification with a defined basal and apical layer and the formation of a zonula adherens-like structure (Supplementary Fig. 1a, b). To investigate the function of α-catenin within cell–cell junctions in a physiologically defined context, we focused on this well-defined process and examined the transition from punctate adhesions (PA) to mature adhesions (MA) in murine epidermal keratinocytes (MEKs).

### Investigating α-catenin in AJs using live-cell FLIM

We used a cell culture protocol for our experiments, in which differentiation of MEKs is induced by the addition of $Ca^{2+}$, leading to the formation and maturation of AJs. We adopted a previously described α-catenin tension sensor (TS) with the initial goal to determine under which circumstances α-catenin becomes exposed to mechanical loads during AJ maturation[15]. This biosensor was originally constructed by inserting our previously reported TS module[16–18], in which a mechanosensitive linker peptide connects a FRET pair of fluorophores, after amino acid (aa) 697 of α-catenin (Fig. 1d). We regenerated this construct by integrating an optimized TS module with increased sensitivity at low piconewton forces, FL-TSM[17], into the same

insertion site (FL-TS-697; Fig. 1d), and also generated a C-terminal fusion construct to control for force-independent effects (FL-TS-C-term). In addition, we cloned our recently described non-stretchable and thus force-insensitive F7 noforce (no-force (NF)) module[19] into the same integration site at aa 697 (Fig. 1d; F7-NF-697); an overview of all here relevant expression constructs is shown in Supplementary Fig. 2.

The advantage of using the F7-NF construct is that both the TS and FRET control are exposed to the exact same microenvironment within the α-catenin molecule, allowing for better control of force-unspecific effects caused by changes in the protein microenvironment. Expression of these constructs in MEKs resulted in the expected localization to AJs (Fig. 1e), while their expression in α-catenin-depleted cells restored the defective localization of vinculin to cell–cell adhesion sites (Fig. 1f), confirming the previously described functionality of the α-catenin molecule after TS insertion[15]. We then performed our established FRET analyses[20,21] by investigating AJs in a mixed population of differentiating MEKs, displaying both PA and MA, using live-cell FLIM. Intriguingly, we observed a strong variation of FRET efficiencies in AJs of FL-TS-697 expressing MEKs, with values ranging from 5–35% (Fig. 1g), a data spread we typically do not encounter in our molecular force measurements[17,22,23]. Although, in theory, this could have been indicative of an unusually wide range of forces acting upon α-catenin, it became apparent that both α-catenin TS and the F7-NF control displayed a similarly broad distribution in FRET efficiencies (Fig. 1g). This suggested that our FLIM-FRET experiments not only detect distinct mechanical states of α-catenin.

### The ABD domain of α-catenin changes conformation during AJ maturation

To identify the underlying cause of the wide range in FRET efficiencies during AJ maturation, we considered the three main factors that can significantly alter the transfer rates of TS constructs: mechanical tension, intermolecular FRET (i.e., energy transfer between neighboring molecules), and conformational changes of the target protein[18,20,24]. To ensure that FRET data were not affected by differences in mechanical forces acting on α-catenin, we focused our further analyses on the force-insensitive F7-NF-697

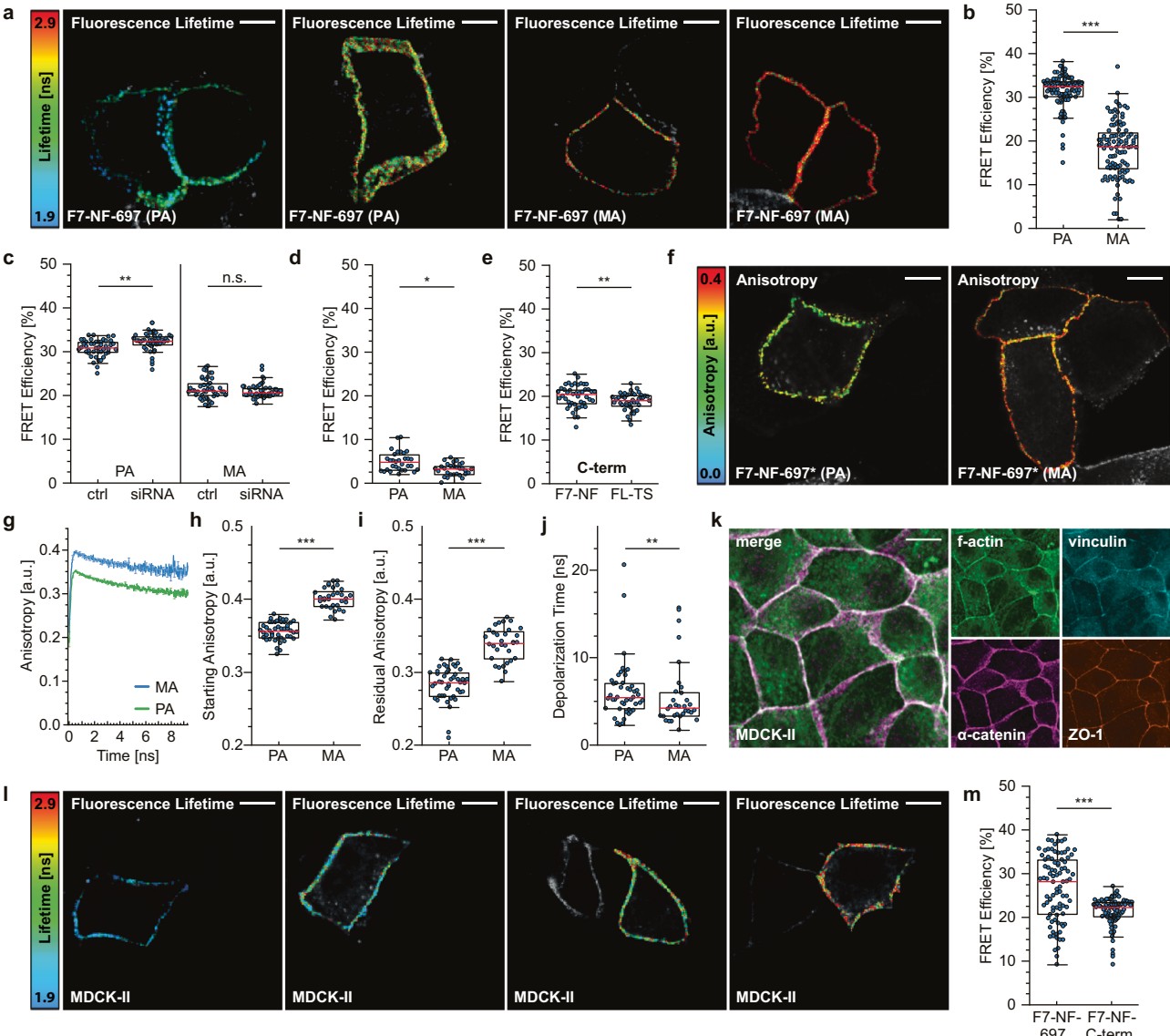

**Fig. 2 | α-catenin undergoes a conformational change during AJ maturation.**
**a** Representative FLIM images of differentiated MEKs expressing the force-insensitive F7-NF-based sensor (F7-NF-697). Note the increased fluorescence lifetimes in MA. **b** Quantification of live-cell FLIM measurements of differentiated MEKs expressing F7-NF-697. PA and MA are characterized by significantly different FRET efficiencies ($N = 6$ replicates, $n = 105, 104$ cells). **c** Live-cell FLIM measurements of differentiated MEKs stably expressing F7-NF-697. Cells were treated with either scrambled (ctrl) or an α-catenin targeting siRNA (siRNA), and classified into PA and MA ($N = 3$ replicates, $n = 45, 45, 45, 45$ cells). **d** FLIM data of differentiated MEKs co-expressing intermolecular FRET controls with either inactivated donor or acceptor chromophore ($N = 3$ replicates, $n = 31, 29$ cells). **e** MEKs expressing α-catenin expression constructs with C-terminal fusions of FL-TS or F7-NF display similar FRET efficiencies ($N = 3$ replicates, $n = 45, 45$ cells).
**f** Representative anisotropy images of differentiated MEKs expressing the force-insensitive F7-NF-based sensor in which the acceptor was inactivated by a point mutation (F7-NF-697*). **g** Mean time-resolved fluorescence anisotropy shown with the standard error of the mean (s.e.m.); data are categorized into PA (green) and MA (blue). **h, i** Quantification of live-cell time-resolved fluorescence anisotropy imaging of differentiated MEKs expressing F7-NF-697*. PA and MA are characterized by different starting (**h**) and residual anisotropies (**i**). **j** The depolarization time is slightly reduced in MA but overall similar to PA. ($N = 3$ replicates, $n = 45, 32$ cells). **k** Representative images of fixed and immunostained MDCK-II cells; merged images display signals from the f-actin and α-catenin channels. **l** Representative FLIM images of MDCK-IIs expressing the force-insensitive F7-NF-based sensor (F7-NF-697). **m** Live-cell FLIM measurements of MDCK-II cells showing comparable variation in F7-NF-697 FRET efficiencies as seen in pooled MEK dataset ($N = 6$ replicates, $n = 90, 90$ cells). Boxplots show the median, the 25th and 75th percentiles and whiskers reaching to the last data point within 1.5× interquartile range. Scale bar: 10 μm. Two-sample KS test: ***$p < 0.001$, **$p < 0.01$, *$p < 0.05$, n.s. (not significant) $p \geq 0.05$.

control and classified the FRET data according to the state of junctional maturation. Notably, data from PA were characterized by high energy transfer rates of about 32% (corresponding to low lifetimes), whereas FRET values were significantly reduced in MA to about 20% as indicated by high lifetimes (Fig. 2a, b). We obtained similar results when cells were treated with para-amino blebbistatin to acutely reduce myosin activity or with Calyculin-A to increase cellular contractility suggesting that the F7-NF-697 construct is rather insensitive to changes in actomyosin tension

(Supplementary Fig. 3). These results were also observed in cells, in which the endogenous α-catenin was depleted by 3'-UTR specific siRNA transfection, showing that the FRET differences in early and mature junctions were not caused by protein overexpression (Fig. 2c and Supplementary Fig. 4a–c). To determine whether the FRET differences between PA and MA were due to variations in intermolecular FRET, we generated two constructs in which the chromophore of either the donor or the acceptor was inactivated by a point mutation (Supplementary Fig. 2). Any FRET observed in

cells co-expressing these constructs results from energy transfer between neighboring molecules. However, we detected low and quite similar FRET efficiencies in PA and MA at around 2–5% (Fig. 2d), suggesting that the stark difference in transfer rates observed before was also not caused by varying intermolecular FRET. We therefore hypothesized that the observed differences in FRET efficiency between PA and MA might be due to a conformational change in the α-catenin molecule, specifically affecting the FRET module when integrated at aa 697. To test this, we analyzed AJs from cells expressing constructs with C-terminal sensor fusions and indeed observed a narrow distribution of FRET efficiencies at about 20% under all conditions (Fig. 2e). This suggested that α-catenin adopts different conformational states in PA and MA, which is specifically detected by the F7-NF construct when inserted into the ABD domain after aa 697.

To confirm these findings with a FRET-independent technique, we generated a construct in which the acceptor fluorophore of the F7-NF-697 module was inactivated by a chromophore point mutation (F7-NF-697*; Supplementary Fig. 2). We reasoned that a significant change in protein conformation should alter how freely the donor fluorophore can rotate within the adhesion structure and that this should be detectable by live-cell fluorescence anisotropy imaging microscopy (FAIM). In this approach, the sample is excited with polarized light and the degree of depolarization, which depends, among other factors, on the rotational diffusion of the excited molecule, is detected and expressed as anisotropy[25]. Indeed, when cells expressing the F7-NF-697* construct were subjected to time-resolved FAIM analysis, significantly different fluorescence anisotropy values were observed in MA when compared to PA (Fig. 2f–i and Supplementary Fig. 5), indicating that the α-catenin molecules adopt different conformational states in the two adhesion structures. Consistent with our previous FLIM-FRET measurements, we detected similar depolarization times in PA and MA, which indicates similarly small contributions of homo-FRET (i.e., intermolecular FRET) to the anisotropy measurement (Fig. 2j). Taken together, these data demonstrate that AJ-resident α-catenin changes its conformational state during AJ differentiation in keratinocytes.

To test whether this conformational change is specific for skin cell differentiation, we expressed the F7-NF-697 module and the C-terminal control in Madin-Darby canine kidney II (MDCK-II) cells, which mimic simple epithelia and form robust AJs (Fig. 2k). Although these cells differ from keratinocytes in that they do not form clearly distinguishable adhesion types that allow a straightforward categorization into PA and MA, we again observed a broad distribution of FRET efficiencies with the conformation-sensitive probe and a narrow distribution in cells expressing the C-terminal, conformation-insensitive control (Fig. 2l, m). This suggests that the conformational change in α-catenin observed here occurs in simple and stratified epithelia.

### The conformation change in the α-catenin ABD correlates with f-actin organization

Given that the FRET module is integrated immediately after the H1 helix (Fig. 1a), we speculated that the observed change in FRET efficiencies may reflect the previously proposed structural change in the ABD of α-catenin[11,13] and would therefore be sensitive to the overall organization of the f-actin cytoskeleton. To explore this further, we measured FRET efficiencies in MEKs expressing the F7-NF-697 construct after modulating f-actin network organization. We started by removing insulin, growth factors and fetal calf serum from the medium to starve the cells, which has been previously shown to block the transition from dot-like to belt-like adhesions in epithelial cells of the mammary gland[26]. As expected, starvation induced a similar phenotype in MEKs, which adopted PA-like adhesions and displayed prominent stress fibers but did not show any cortical actin organization; overall, the cellular phenotype was reminiscent of early stages of AJ formation (Fig. 3a). Indeed, under these conditions, we observed energy transfer rates similar to those observed in the PA of untreated MEKs of about 30% (Fig. 3b). Next, we treated cells with the actin polymerization inhibitor Cytochalasin-D (CytoD) or with

DMSO alone, as a control. Addition of CytoD led to actin network disassembly and the loss of cortical actin organization, an effect that was reversible by CytoD wash-out, while the addition of DMSO did not visibly impact the cell adhesion structures (Fig. 3c, d and Supplementary Fig. 6a, b). FRET efficiencies in the resulting PA-like adhesions were again characterized by high FRET efficiencies at approximately 30% (Fig. 3e). As expected, starvation or CytoD treatment did not impact FRET values in C-terminally tagged α-catenin controls, confirming that the effects are exclusively detected when the FRET module was inserted within the molecule, C-terminal of the H1 helix (Fig. 3f). To mimic an actin network organization characteristic of late maturation stages, we stabilized microtubules by Taxol-treatment, which has been demonstrated to promote a cortical actin organization[27]. Consistent with these reports, we observed a MA-like phenotype with prominent actin fibers at the cortex, a phenotype that persisted for hours after Taxol wash-out (Fig. 3g and Supplementary Fig. 6c). Under these circumstances, FRET efficiencies in F7-NF-697 expressing cells were low and indistinguishable from those observed in MA of untreated cells (Fig. 3h); C-terminal controls remained virtually unchained after Taxol-treatment with a narrow FRET efficiency distribution at around 20% (Fig. 3i). We infer from these experiments that the α-catenin conformation change monitored with the F7-NF-697 construct correlates with actin network organization.

### The conformational switch in the ABD is distinct from structural rearrangement in the MD

Earlier studies have shown that α-catenin undergoes a force-dependent, conformational change in its MD, ultimately leading to vinculin recruitment and cell adhesion reinforcement[4–6]. To test to which extent the previously described conformational change in the MD[8] reflects the structural transition described here, we reproduced the α-catenin MD conformation sensor (MD-CS) by inserting a donor fluorophore (mTurquoise2) after aa 315 of α-catenin and an acceptor fluorophore (ShadowG) after aa 639. To allow a quantitative analysis and determine FRET efficiencies, we also established a donor-only control by mutating the acceptor chromophore (Fig. 4a and Supplementary Fig. 2). We expressed these constructs in wild type (wt) or α-catenin-depleted MEKs and confirmed the expected localization to mature AJs (Fig. 4b, c), as described before[8]. In contrast to the F7-NF-697-based experiments, however, the MD-CS expressing cells displayed a comparably low and narrow distribution of FRET efficiencies at around 13% in a mixed population of cells. Since the MD-CS construct poorly localized to PA, we treated cells with CytoD (or Taxol) to alter actin network organization, which had induced high (or low) FRET efficiencies in the F7-NF-697 construct (Fig. 3). However, these treatments did not significantly alter transfer rates in AJs of MD-CS expressing cells (Fig. 4d), demonstrating that the conformation change during AJ maturation observed with the F7-NF-697 sensor is distinct from the previously reported conformational change in α-catenin's MD.

The structural rearrangement in the MD domain allows the force-dependent recruitment of vinculin to α-catenin, leading to an enhanced association with the actin cytoskeleton[4–6]. To investigate whether direct vinculin binding is required for the conformational transition in the ABD, we introduced mutations into α-catenin that abolish (L344P) or enhance (M319G/R326E) vinculin binding[7,28] (Fig. 4e). Incorporation of the mutations into the F7-NF-697 construct and expression in MEKs did not visibly affect localization to AJs (Fig. 4f) and allowed for normal PA initiation and maturation to MA. Evaluation by live-cell FLIM, however, revealed no significant differences in transfer rates between cells expressing wt and mutant α-catenin constructs, showing high FRET in PAs and reduced values in MA for all constructs (Fig. 4g). Thus, the structural change in the ABD of α-catenin during AJ maturation does not require direct vinculin binding to α-catenin and may occur prior to or independently of force-dependent vinculin recruitment.

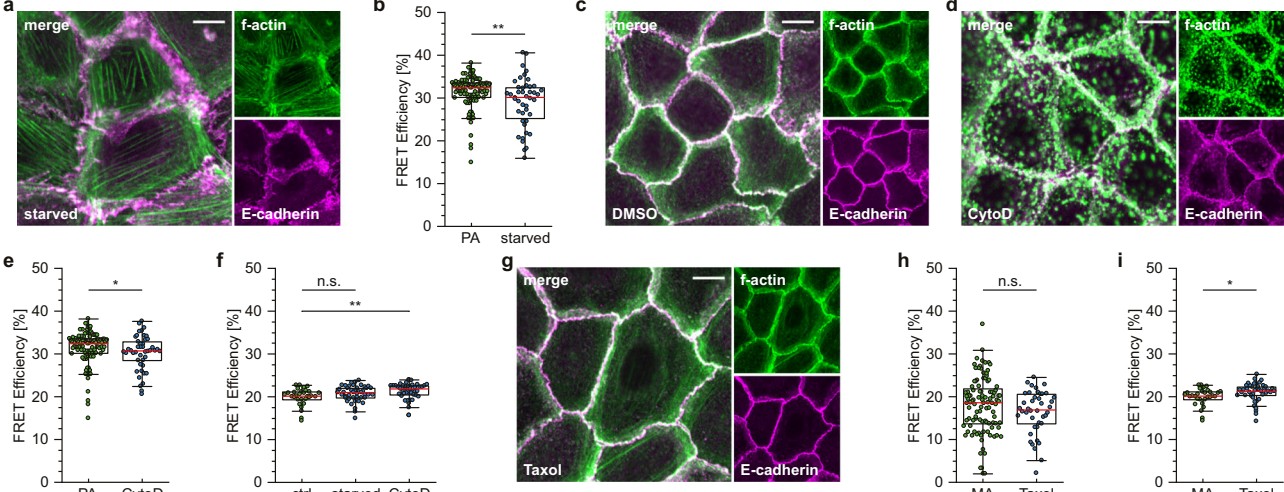

Fig. 3 | The conformational change in α-catenin is sensitive to actin network organization. a Representative image of differentiated MEKs immunostained after 24 h of serum starvation. Actin fibers (green) are connected to clusters of E-cadherin (magenta). b Live-cell FLIM data of differentiated and serum-starved MEKs expressing F7-NF-697. Data from the starved condition (blue) are shown next to data from PA under untreated conditions (green, same data as Fig. 2b) (green $N = 6$ replicates, $n = 105$; blue $N = 3$ replicates, $n = 44$). c Representative image of differentiated MEKs immunostained after 1 h of DMSO treatment. d Representative image of differentiated MEKs immunostained after 1 h of CytoD treatment (2 μM); cells show an impaired cortical actin network (green) and punctate E-cadherin signals (magenta). e Live-cell FLIM data of differentiated and CytoD-treated MEKs. For comparison, data from CytoD-treated cells (blue) are shown next to data from PA under untreated conditions (green, same data as Fig. 2b) (green $N = 6$ replicates, $n = 105$; blue $N = 3$ replicates, $n = 45$). f Live-cell FLIM data of differentiated MEKs expressing α-catenin controls with a C-terminal sensor fusion. Serum starvation and

CytoD treatment (blue) have negligible effects on the overall FRET efficiency as compared to untreated cells (green) ($N = 3$ replicates, green $n = 30$ cells; blue $n = 44$, 45 cells). g Representative image of differentiated MEKs immunostained after 1 h of Taxol-treatment (10 μM). A distinctive cortical f-actin localization is accompanied by an augmented junctional E-cadherin signal. h Live-cell FLIM data of differentiated and Taxol-treated MEKs expressing F7-NF-697. Data from Taxol-treated cells (blue) are shown next to data from MA under untreated conditions (green, same data as Fig. 2b) (green $N = 6$ replicates, $n = 104$ cells; blue $N = 3$ replicates, $n = 42$ cells). i Live-cell FLIM quantifications of differentiated MEKs expressing an α-catenin control with a C-terminal sensor fusion. Taxol-treatment (blue) has negligible effects on the overall FRET efficiency when compared to untreated cells (green) ($N = 3$ replicates; green: $n = 30$ cells; blue $n = 42$ cells). Boxplots show the median, the 25th and 75th percentiles and whiskers reaching to the last data point within 1.5× interquartile range. Scale bar: 10 μm. Two-sample KS test: **$p < 0.01$, *$p < 0.05$, n.s. (not significant) $p \geq 0.05$.

## The conformation switch in the ABD correlates with altered turnover rates

The ability of cell adhesion structures to withstand mechanical stresses and simultaneously enable dynamic, cellular responses is thought to originate from a constantly ongoing protein turnover of the resident cell adhesion proteins. In particular, cytoplasmic molecules linking cell adhesion receptors to the actin cytoskeleton—like focal adhesion proteins in cell-matrix adhesions[29] or catenins in AJs[6,30,31]—typically exchange within tens of seconds. We therefore wanted to determine how the apparent change in the ABD of α-catenin, presumably enforcing a stabilized, catch-bond state, correlates with the turnover rate of the molecule in cell–cell junctions.

To perform these experiments, we used the F7-NF-697 sensor to record α-catenin conformation in PA and MA by FLIM-FRET and then performed a fluorescence recovery after photobleaching (FRAP) experiment to determine molecular turnover. Consistent with a previous study[6], we observed a rather wide range of mobile fraction values when data from a mixed population of cells with PA and MA were analyzed. However, categorizing the data into PA and MA yielded more differentiated results in which protein turnover seemed to change with the state of junction maturation (Fig. 5a–c). To exclude non-specific effects on the fluorescence intensity measurements by FRET, we repeated the experiments using donor-only controls in which the acceptor chromophore was mutated, but these results were highly similar to those obtained in cells expressing the intact FRET construct indicating distinct protein dynamics in PA and MA (Supplementary Fig. 7). In both cases, the presumably stabilized MA were associated with faster protein turnover rates, whereas PA displayed slower recoveries (Fig. 5c, d), and the decrease in FRET efficiency linearly correlated with an increase in mobile fraction (Fig. 5e). As expected, and consistent with the notion that α-catenin turnover is actin network dependent, we observed a strongly reduced α-catenin recovery after CytoD treatment. To

reduce the effects of photobleaching from the preceding FLIM-FRET experiments, we complemented these data sets with standard FRAP experiments in which AJ dynamics were monitored for up to 20 min. These experiments again showed a significantly higher protein turnover in MA compared to PA (Fig. 5f). As the recovery curves could be described very well by bi-exponential fits, the data suggest that at least two different pools of α-catenin contribute to the observed protein turnover in both PA and MA (Fig. 5f).

Altogether, the results show that the conformational change in the ABD of α-catenin directly correlates with the subcellular dynamics of the AJ-resident molecules. Unexpectedly, the data reveal that the overall population of α-catenin molecules can be turned over efficiently in mature cell–cell junctions.

## Discussion

The assembly and function of mammalian cell adhesion structures are critically dependent on the regulated engagement of adhesion receptors with the cytoskeleton. It is widely recognized that the linkage between cadherin receptors and the actin network is mediated by α-catenin, which binds actin directly through its C-terminal ABD[31,32]. Previous studies have suggested that this interaction is accompanied by a conformational change in α-catenin's ABD to allow for a high-affinity interaction that can withstand physiologically relevant mechanical stresses[11,13,33]. However, where and when α-catenin undergoes the structural transition at its C-terminal domain has remained unclear because suitable tools to study the effect in the physiological context of the living cell were missing. With this study, we introduce a genetically encoded α-catenin sensor to visualize and quantify the ABD-associated conformational change in cell–cell adhesions of living cells using either fluorescence lifetime or anisotropy imaging. Given the remarkably robust FRET efficiency difference of about 10% between the

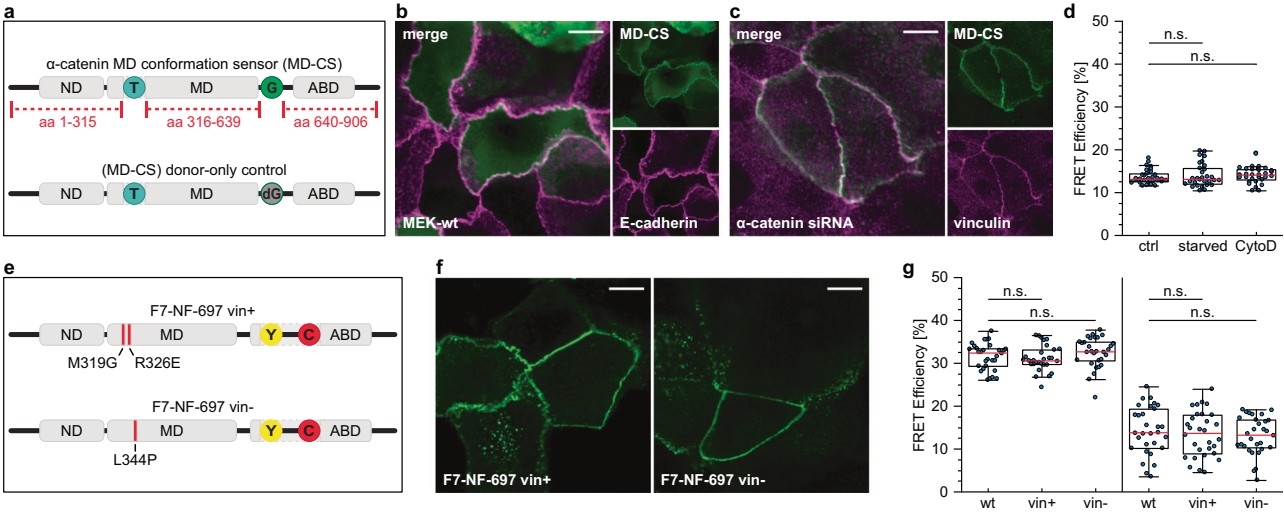

**Fig. 4 | The structural change in the ABD is distinct from the MD conformational change. a** Schematic depiction of the α-catenin MD conformation sensor (MD-CS)[8], regenerated in this study. As a donor fluorophore, mTurquoise2 (T) was inserted after aa 315; ShadowG (G) was used as an acceptor and inserted after aa 639. A donor-only control was generated by point-mutating (Y67G) ShadowG. **b** Representative image of differentiated MEKs expressing the MD-CS construct (green), stained for E-cadherin (magenta), showing efficient recruitment of the biosensor to MA. **c** Representative image of α-catenin-depleted MEKs expressing the MD-CS construct (green), fixed and stained for vinculin (magenta). Note the enhanced vinculin recruitment in cells rescued by expression of the MD-CS. **d** Quantification of live-cell FLIM experiments in differentiated MEKs expressing MD-CS under untreated (ctrl), CytoD- and Taxol-treated conditions (N = 2 replicates, n = 30, 30, 30 cells). **e** Schematic depiction of the vinculin-

binding mutants inserted into F7-NF-697. The double point-mutation M319G/R326E (vin +) has been reported to allow vinculin binding to α-catenin in a force-independent manner; the mutation L344P (vin-) impairs vinculin binding. **f** Representative images of differentiated MEKs expressing either F7-NF-697 vin+ (left) or F7-NF-697 vin- (right) showing normal biosensor recruitment to AJs. **g** Quantification of live-cell FLIM experiments of differentiated MEKs expressing either wt, vin +, or vin- F7-NF-697. Data were categorized into PA and MA, showing that the conformational change is insensitive to direct vinculin binding (N = 3 replicates, n = 30, 30, 30, 30, 30, 30 cells). Boxplots show the median, the 25th and 75th percentiles and whiskers reaching to the last data point within 1.5× interquartile range. Scale bar: 10 μm. Two-sample KS test: n.s. (not significant) p ≥ 0.05.

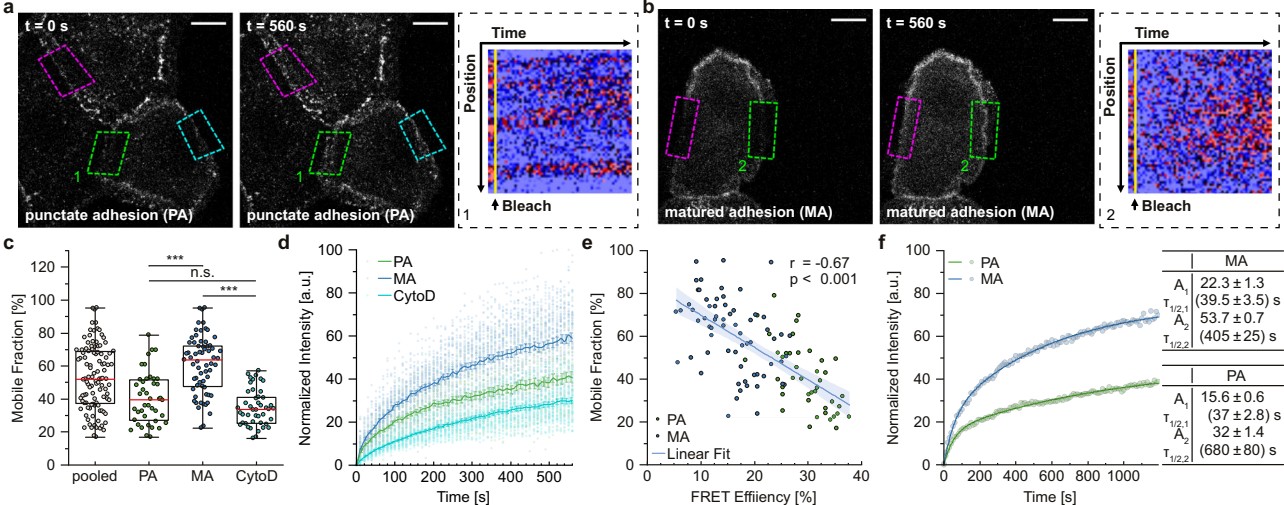

**Fig. 5 | The conformational change in the ABD correlates with an increased α-catenin turnover. a, b** Representative intensity images of differentiated MEKs expressing F7-NF-697 in PA (**a**) and MA (**b**) directly after photobleaching and after 560 s of fluorescence recovery. Dashed squares in magenta, cyan, and green indicate photobleached regions of interest. Kymograms show fluorescence recovery from regions 1 (PA) and 2 (MA) in the green dashed squares. **c** The calculated mobile fractions indicate a faster protein turnover of α-catenin molecules in MA. Low mobile fractions are associated with PA and can be further dampened by CytoD treatment (pooled, PA and MA: N = 5 replicates, n = 107, 42, 65 junctions; CytoD: N = 3 replicates, n = 45 junctions). **d** Recovery of the normalized mean fluorescence intensities shown with s.e.m. from FRET-FLIM/FRAP experiments in cells expressing F7-NF-697; data are categorized into PA (green) and MA (blue). Data

from cells treated with CytoD are shown in cyan. **e** Quantification of live-cell FRET-FLIM/FRAP experiments in MEKs expressing F7-NF-697. Plotting mobile fractions over FRET efficiencies obtained from individual AJs reveals a correlation between α-catenin conformation and AJ turnover. The regression line, with a 95% confidence interval, was calculated by a Spearman correlation test (n = 107 values). **f** Regression curves obtained from bi-exponential fitting of data from standard FRAP experiments; data are categorized into PA (green) and MA (blue). Fitting parameters are shown in tables; values are given with standard error (N = 6 replicates, n = 40, 40). Boxplots show the median, the 25th and 75th percentiles and whiskers reaching to the last data point within 1.5× interquartile range. Scale bar: 10 μm. Two-sample KS test: ***p < 0.001, n.s. (not significant) p ≥ 0.05.

switched and non-switched conformations of α-catenin, we expect the probe to be broadly applicable for evaluating the state of the α-catenin–actin linkage in vivo. It should be particularly interesting to apply the tools developed here to α-catenin in fruit flies or zebrafish, where the molecule plays a critical role in the maintenance and mechanical integrity of cell–cell adhesions in various tissues, such as the imaginal disc or follicle cells in flies and the enveloping layer in fish[34,35]. Since both model systems are amenable to high-resolution fluorescence imaging such as FRET microscopy, and since α-catenin is highly conserved at the amino acid sequence level between these species—especially at the insertion site used here—the experiments seem feasible and should provide direct insights into the role of the ABD switch in vivo.

Our first cell culture experiments reveal that α-catenin adopts at least two different C-terminal conformations in AJs, which occur during epidermal differentiation, but are also evident in simple epithelia. In the keratinocytes used in this study, the ratio in which α-catenin occupies the two states depends on the degree of junctional maturation and correlates with the organization of the F-actin cytoskeleton. Consistent with previous descriptions of junction maturation[36,37], showing the early formation of punctate structures that gradually reorganize into linear, mature adhesions, AJs in our cells changed their appearance from PA to MA and were characterized by progressively reduced FRET efficiencies over the course of hours. When cells adopted a uniform cortical actin network with a presumably coordinated, directional actin flow[38], strongly reduced FRET values were observed, suggesting that the majority of α-catenin molecules had altered their ABD conformation. Our vinculin mutant experiments suggest that the structural change at the ABD is rather insensitive to vinculin binding, which is thought to modulate junction maturation by engaging the MD domain[37]. Notwithstanding potential cell type-specific differences, these findings suggest that both conformational changes—the structural rearrangement in the MD, which is tension sensitive and facilitates vinculin recruitment, and the structural rearrangement in the ABD—both contribute to AJ maturation, but can occur independently.

These data also seem consistent with previous force spectroscopy measurements showing the formation of a catch-bond interaction between actin and α-catenin's ABD, which is strictly dependent on the orientation of the applied stress[11,12], and it is tempting to speculate that the biosensor developed here specifically monitors the described α-catenin–actin catch-bond state. Yet, establishing further unambiguous evidence that F7-NF-697 is a catch-bond sensor is currently challenging due to the lack of a conformation-insensitive α-catenin force probe. Such a biosensor could be used to correlate molecular tension across α-catenin with ABD conformation; however, the previously developed α-catenin tension sensor[15] seems unsuitable in our system to delineate force-specific from conformation-dependent effects (Figs. 1 and 2). It will therefore be important to further optimize existing probes to facilitate more specific α-catenin tension measurements.

Interestingly, the different conformational states of α-catenin correlate with protein turnover in a seemingly counterintuitive way: mature adhesions, with presumably strongly bound α-catenin–actin linkages, have higher mobile fractions compared to early, punctate adhesions, resulting in a mechanically resilient but still dynamic adhesion structure. This finding suggests that not all α-catenin molecules within a stabilized junction are engaged in the actin tethering state. A significant fraction of molecules must be available for protein turnover and may act as bystanders that are exposed to negligible mechanical stresses[12,13]. This may be possible because mechanical loads can also be borne by alternative force-bearing linkages, for example, through myosin VI[39] or β-catenin, which has recently been proposed to reinforce AJs through direct vinculin binding[40]. Thus, it appears that mechanical loads are distributed in mature AJs across a number of force-bearing linkages, allowing for efficient turnover of unloaded α-catenin molecules.

Our FRAP analysis suggests that the population of α-catenin molecules in early and mature adhesions is replenished from different subcellular pools, resulting in a higher mobile fraction in MA (Fig. 6). The data do not suggest that apical, membrane-associated α-catenin plays an important role in this process, as bleached areas show uniform fluorescence recovery. In contrast, MA may benefit from a previously described subcellular exchange of adhesion molecules between basal and apical adhesion structures[41]. Irrespective of the underlying mechanisms, the experiments emphasize that the conformational change in α-catenin's ABD is central to the formation of an adhesion structure that is mechanically resilient but inherently dynamic.

**Fig. 6 | Conformation of α-catenin's ABD and protein dynamics during AJ maturation.** The C-terminus of α-catenin changes conformation during junction maturation, which can be monitored with the FRET-based conformation sensor introduced here. The population of α-catenin molecules in punctate and mature adhesions is characterized by distinct turnover dynamics. The data indicate that a fraction of α-catenin molecules is exchanged with the cytoplasmic pool on the second time scale, while other molecules may be exchanged on the order of tens of seconds through alternative routes. As described previously[39], a direct exchange between punctate adhesions at the basal side of cells and mature adhesions at apical regions could result in higher turnover rates at mature adhesions. Cyan arrows indicate turnovers suggested by our FRAP experiments, gray arrows indicate exchange processes that were not analyzed here but are likely to exist. Note that alternative, potential force-bearing linkages, for example, through direct β-catenin-vinculin binding[40], are not shown.

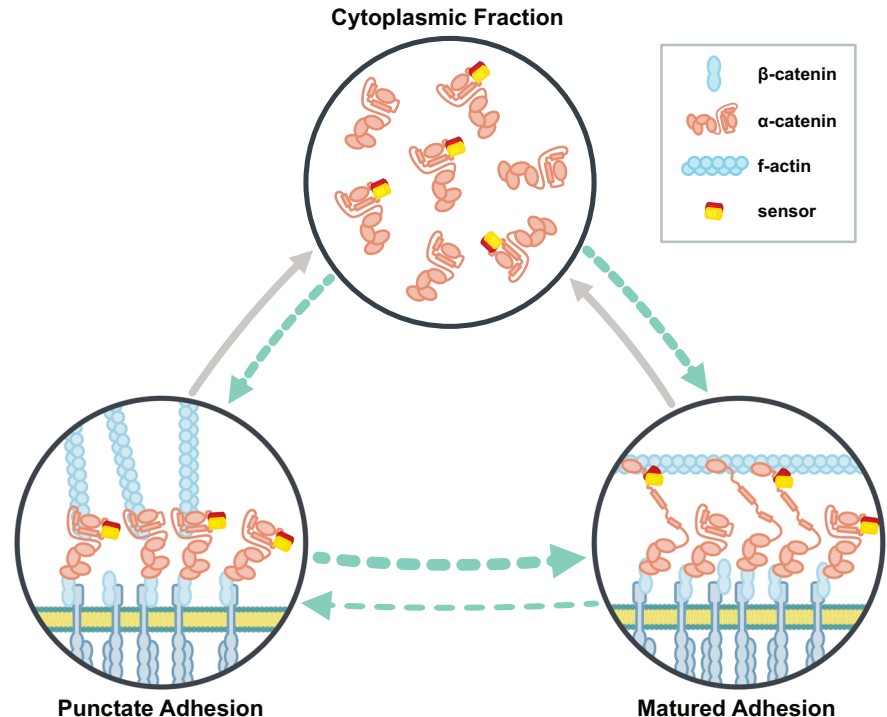

The genetically encoded tools developed in this study should facilitate the investigation of where and when this process occurs under physiologically relevant conditions in living cells and model organisms.

## Methods

### Antibodies and reagents

The following primary antibodies were used at the indicated dilutions for immunofluorescence staining (IF) and Western blotting (WB): rabbit anti-α-catenin (Sigma-Aldrich, C2081; IF: 1/1000; WB: 1/7500), rat anti-E-cadherin (Thermo Fisher Scientific, 13-1900; IF: 1/1000), rabbit anti-phospho-myosin light chain 2 (Thr18/Ser19) (Cell Signaling, 3674; IF: 1/200), mouse anti-tubulin (DM1A) (Sigma-Aldrich, T6199; WB: 1/10,000), mouse anti-vinculin (Sigma-Aldrich, V9131; IF: 1/400), rat anti-ZO-1 (R26.4C) (Thermo Fisher Scientific, 14-9776-82; IF: 1/100). The following secondary antibodies were used at the indicated dilutions: anti-mouse immunoglobulin G (IgG) Alexa Fluor 405 (Thermo Fisher Scientific, A31553; IF: 1/200), anti-mouse IgG Alexa Fluor 488 (Thermo Fisher Scientific, A11001; IF: 1/500), anti-mouse IgG HRP (Bio-Rad, 170-6516; WB: 1/10000), anti-rabbit IgG Alexa Fluor 568 (Thermo Fisher Scientific, A11036; IF: 1/500), anti-rabbit IgG Alexa Fluor 647 (Thermo Fisher Scientific, A21244; IF: 1/500), anti-rabbit IgG HRP (Bio-Rad, 170-6515; WB: 1/10,000), anti-rat IgG Alexa Fluor 488 (Thermo Fisher Scientific, A11006; IF: 1/500), anti-rat IgG Alexa Fluor 568 (Thermo Fisher Scientific, A11077; IF: 1/500). Alexa Fluor 568 Phalloidin (Thermo Fisher Scientific, A12380; IF: 1/200) was used to visualize f-actin. The following chemicals were used at the indicated concentrations and time: Taxol (Sigma-Aldrich, T7402; 10 μM; 1 h), Cytochalasin-D (Sigma-Aldrich, C8273; 2 μM; 1 h), para-amino-blebbistatin (Sigma, AA0200ZV; 50 μM; 1 h), Calyculin-A (Cell Signaling, 9902S; 10 nM; 1 h). The following siRNAs were used for knockdown experiments: custom-designed siRNA against 3'UTR of mouse α-catenin 5'-CAUUAAUGGAGUUGCUUUAdTdT-3' (Eurofins), non-specific control 31% GC 5'-UAAUGUAUUGGAACGCAUAdTdT-3' (Eurofins).

### Generation of expression constructs

Our previously published protocols[21], using NEBuilder HiFi DNA Assembly Master Mix (New England Biolabs, E2621L), were used to generate all expression constructs in this study. For all α-catenin constructs aCat TL1 (1-907 mTFP1-TSmod-Venus STOP) (Addgene #101297) was used as a template. Donor-only and intermolecular FRET controls were generated by Gibson assembly and used fluorescence-inactivated mCherry(Y72L), fluorescence-inactivated YPet (Y67G) or non-quenching ShadowG (Y67G). Final cDNA sequences were assembled in the expression vector pLPCX (Clontech, 631511). Mutations in α-catenin (M319G, R326E) and α-catenin (L344P)[42] were also created using the Gibson assembly approach. The correct cDNA sequences were confirmed by DNA sequencing (Eurofins, Microsynth Seqlab).

### Cell culture conditions

Mouse Epidermal Keratinocytes (MEKs) were cultured at 32 °C and 5% $CO_2$ in complete FAD medium: DMEM/Ham's F12 3.5:1.1, low-calcium (0.05 mM Ca2 +) (custom-made by PANBiotech) supplemented with 10% Chelex-treated FBS, 2 mM GlutaMAX, 1 mM sodium pyruvate, 0.18 mM adenine, 0.5 μg/ml hydrocortisone, 5 μg/ml insulin, 10 ng/ml EGF, 100 pM cholera toxin, and 0.5% P/S. MEKs were seeded on collagen I-coated plasticware. To induce formation of cell–cell junctions, complete FAD medium was supplemented with 1.2 mM $Ca^{2+}$ (FAD +) and cells were differentiated for 24 h. For starvation experiments, cells were differentiated for 24 h in DMEM/Ham's F12 supplemented with 2 mM GlutaMAX, 1 mM sodium pyruvate, 0.5% P/S and 1.2 mM $Ca^{2+}$. For live-cell imaging, 25 mM HEPES was added to the differentiation medium. For experiments, approximately $5 \times 10^4$ cells/cm$^2$ were seeded on collagen I-coated imaging dishes (ibidi, 81156) 24 h prior to Ca$^{2+}$-switch. cDNA constructs were expressed by transient transfection using 2.5 μg DNA, 5 μl P3000 reagent and 3.75 μl Lipofectamine 3000 for 4 h according

to the manufacturer's protocol before Ca$^{2+}$-switch. siRNA-mediated knockdown was achieved by transfection using 25 pmol siRNA and 7.5 μL Lipofectamine RNAiMAX (Thermo Fisher Scientific, 13778500) 48 h prior to Ca$^{2+}$-switch. Co-transfections of DNA and siRNA were performed by using 2.5 μg DNA, 25 pmol siRNA and 3.75 μl Lipofectamine 3000 according to the manufacturer's protocol 48 h before Ca$^{2+}$-switch. Stable cell lines were established using the Phoenix cell transfection system. Ecotropic, retroviral particles were produced according to established protocols[43] with slight adjustments: retrovirus was collected after 48 h of incubation at 32 °C and concentrated using Retro-X Concentrator (Takara, 631455) according to the manufacturer's protocol and resuspended in complete FAD. MEKs were seeded at a density of $4.5 \times 10^3$ cells/cm$^2$ 24 h prior to transduction. Cells were transduced for 24 h in the presence of polybrene (5 μg/ml). 48 h after transduction, cells were passaged and selected using 2 μg/ml puromycin. A population of highly expressing MEKs was achieved by FACS. MDCK-II cells were cultured at 37 °C and 5% $CO_2$ in DMEM supplemented with 5% FBS and 1% P/S. For experiments, MDCK-II cells were seeded at a density of approximately $3.5 \times 10^4$ cells/cm$^2$ on imaging dishes 24 h prior to transfection. cDNA constructs were expressed by transient transfection, and cells were analyzed the next day.

### Immunostaining

For immunostainings, cells were fixed with 3.5% paraformaldehyde for 15 min at RT and washed in PBS. Samples were blocked for 1 h at RT in PBS containing 2% BSA and 0.1% TritonX-100 (blocking buffer). Antibodies were diluted in blocking buffer, and samples were incubated with primary antibody for 2 h at RT, followed by secondary antibody for 1 h at RT. Image acquisition was carried out on an LSM880 confocal laser scanning microscope (Zeiss) using ZEN Software (black edition, Zeiss), equipped with a 63× immersion objective (LD LCI Plan-Apochromat 63×/1.2 Imm Corr DIC M27).

### Fluorescence recovery after photobleaching (FRAP)

To characterize AJ dynamics, FRAP experiments were performed on differentiated cells. Image acquisition was carried out on an LSM880 confocal laser scanning microscope (Zeiss) using ZEN Software, black edition (Zeiss), equipped with a 63× immersion objective (LD LCI Plan-Apochromat 63×/1.2 Imm Corr DIC M27) and a 32 °C heating chamber. Three pre-bleached images were recorded at 514 nm followed by bleaching of a selected AJ with 100% laser intensity for 80 iterations. Fluorescence recovery was recorded for either 560 s or 1190 s at 10 s intervals. For data analysis, the bleached region was corrected for background and compared with an unbleached control AJ to correct for gradual bleaching during image acquisition. Raw fluorescence recovery curves were extracted by FIJI software and Jay_Plugins (https://research.stowers.org/imagejplugins/zipped_plugins.html); normalized FRAP curves were plotted with OriginPro and fitted with the function (Eq. 1):

$$f(t) = \sum^{n} a_n \left(1 - e^{-b_n t}\right) \tag{1}$$

where (*t*) is time, (*b*) the rate constant and (*a*) the mobile fraction. The recovery half-time ($\tau_{1/2}$) (Eq.2) is defined as:

$$\tau_{1/2,n} = \frac{\ln(2)}{b_n} \tag{2}$$

### Fluorescence lifetime imaging microscopy (FLIM)

FLIM experiments were performed on a LSM880 confocal laser scanning microscope (Zeiss) with integrated PicoQuant FLIM using ZEN Software, black edition (Zeiss) and SymPhoTime 64 software (PicoQuant), equipped with two pulsed lasers for excitation at 440 nm (LDH-D-C-440, 40 MHz

repetition rate) and at 510 nm (LDH-D-C-510, 40 MHz repetition rate), a dichroic mirror (Semrock, 482/35 BrightLine HC), a band-pass filter for mTurquoise2 (Chroma, 480/30 ET), a band-pass filter for YPet (Chroma, 540/30 ET), a FLIM module (MultiHarp 150 4 N), a 63× immersion objective (LD LCI Plan-Apochromat 63×/1.2 Imm Corr DIC M27) and a 32 °C heating chamber. AJ images were acquired over 61.44 x 61.44 μm area (512 × 512 pixels). For each experimental condition, 10–20 images were taken on 2–6 individual days.

## FLIM-FRET analysis

Analysis of FLIM data was performed using SymPhoTime 64 (PicoQuant) as described before[44]. In brief, images were thresholded to exclude background signal and ROIs around AJ were drawn manually. The fluorescence lifetime was determined by use of the "n-Exponential Tailfit" function, which was set to mono-exponential for donor-only experiments and bi-exponential otherwise. In bi-exponential fitting, $\tau_1$ was fixed and set to the median donor-only lifetime. The FRET efficiency ($E_{FRET}$) (Eq. 3) was calculated from the lifetime of the donor in the presence of an acceptor ($\tau_{DA}$) (Eq. 4) and the median donor-only lifetime ($\tilde{\tau}_D$) according to

$$E_{FRET} = 1 - \frac{\tau_{DA}}{\tilde{\tau}_D} \quad (3)$$

where $\tau_{DA}$ is defined as the average lifetime ($\tau$) weighted by the contributing amplitudes ($A$)

$$\tau_{DA} = \frac{1}{A_{Sum}} \sum_{k=0}^{n-1} A_k \tau_k \quad (4)$$

## Time-resolved fluorescence anisotropy imaging microscopy (tr-FAIM)

Time-resolved fluorescence anisotropy experiments were performed on a LSM880 confocal laser scanning microscope (Zeiss) with integrated Pico-Quant FLIM and a polarization extension unit (PicoQuant) using ZEN Software, black edition (Zeiss) and SymPhoTime 64 software (PicoQuant), equipped with a pulsed laser for excitation at 510 nm (LDH-D-C-510, 20 MHz repetition rate), a band-pass filter for each light polarization path for ATTO 514 (ATTO-TEC, AS 514-21) and YPet (Semrock, 550/49 BrightLine HC), a FLIM module (TimeHarp 260 PICO Dual), a 10× air objective (Plan-Apochromat 10x/0,45 M27), a 63× immersion objective (LD LCI Plan-Apochromat 63×/1.2 Imm Corr DIC M27) and a 32 °C heating chamber.

For setup validation, an ATTO 514 solution (5 μM) containing 0%, 5%, 50%, 60%, 70%, and 80% of glycerol was used. tr-FAIM images of solutions were acquired over ten cycles with a pixel dwell time of 1.26 μs over 386.56 x 386.56.44 μm area (512 × 512 pixels) (10×) and over 61.44 x 61.44 μm area (512 × 512 pixels) (63×). Each ATTO 514 solution was repeatedly probed three times. Fluorescence intensity decays were exported as ASCII files using SymPhoTime 64 (PicoQuant) and processed with OriginPro.

With the 0% glycerol solution, the G-factor, which accounts for the difference of transmission efficiencies of the optics for perpendicular and parallel polarized light, was calculated for the 10× and the 63× objective according to

$$G(t) = \frac{I_{\perp}(t)}{I_{\parallel}(t)} \quad (5)$$

where $I_{\perp}(t)$ is the intensity of perpendicular polarized light and $I_{\parallel}(t)$ the intensity of parallel polarized light.

Final G-factors were obtained by taking the average value from 600 ps to 11.675 ns after excitation, in which the upper limit is thrice the fluorescence lifetime of ATTO 514 measured with our setup (3.89 ns), resulting in $G^{obj10} = 1.29$ and $G^{obj63} = 1.41$ ($n = 444$ time gates). Fluorescence

anisotropy was then determined according to

$$r(t) = \frac{GI_{\parallel}(t) - I_{\perp}(t)}{GI_{\parallel}(t) + x_{NA}I_{\perp}(t)} \quad (6)$$

where $x_{NA}$ is an empirical correction factor that depends on the numerical aperture of the objective. For objectives with low numerical apertures ($< 0.9$), negligible depolarization is induced and $x_{NA} = 2$. Determination of the correction factor for high numerical aperture objectives was based on the measurements of the ATTO 514 solution containing 80% glycerol, and the factor was calculated according to

$$x_{NA}(t) = \frac{GI_{\parallel}^{obj63}(t)\left(1 - r^{obj10}(t)\right) - I_{\perp}^{obj63}(t)}{r^{obj10}(t)I_{\perp}^{obj63}(t)} \quad (7)$$

based on[25]. The final correction factor was then obtained by taking the average value from 600 ps to 11.675 ns after excitation, resulting in $x_{NA} = 1.22$.

AJ images were acquired with the 63×objective over 61.44 x 61.44 μm area (512 × 512 pixels) with a pixel dwell time of 1.26 μs over 60 cycles. For each experiment experimental condition, 9–16 images were taken on 3 individual days.

## Evaluation of tr-FAIM data

Analysis of tr-FAIM data was performed using SymPhoTime 64 (Pico-Quant) and OriginPro. First, images were thresholded to exclude background signal, then ROIs around AJ were drawn manually. Afterwards, resulting intensity decays were exported as ASCII files and processed with OriginPro. Time-resolved fluorescence anisotropy was calculated following Eq. 6 and the determined values for $G^{obj63}$ and $x_{NA}$. Resulting anisotropies were fit from 600 ps to 9.3 ns (thrice the lifetime of YPet in our setup) after excitation according to

$$r(t) = (r_0 - C)e^{-t/\phi} + C \quad (8)$$

where $r_0$ is the starting anisotropy, $\phi$ the depolarization time and $C$ the residual anisotropy accounting for a hindered rotation of fluorophores when integrated into α-catenin. Representative steady-state anisotropy images were generated using the 'Anisotropy Image' function in SymPhoTime 64, in which the binning was set to '2 points', background was removed by thresholding and ROIs around AJs were manually drawn.

## Western blotting

siRNA-treated, differentiated cells were washed with PBS and lysed in M-PER Mammalian Protein Extraction Reagent (Thermo Fisher Scientific, 78501) containing a protease inhibitor cocktail (cOmplete ULTRA, mini, EDTA-free EASYpack, Roche, 5892791001). SDS-PAGE and Western blotting were performed according to standard procedures.

## Statistics and reproducibility

All experiments were conducted in at least three independent experiments, if not stated otherwise, and only reproducible experiments are reported. The number of replicates is indicated in the figure legends as 'N', the sample size as 'n'. FLIM, FAIM and FRAP data are plotted in boxplots generated using OriginPro, showing the median, the 25th and 75th percentiles and whiskers reaching to the last data point within 1.5× interquartile range. To compare statistical significance, a two-sided Kolmogorov–Smirnov (KS) test with a default significance level of $\alpha = 0.05$ was used. Statistical significances are given by the $p$-value: ***$p < 0.001$; **$p < 0.01$; *$p < 0.05$; n.s. (not significant), $p \geq 0.05$. Linear regression was performed with OriginPro, and correlation was calculated by the Spearman rank correlation test.

## Reporting summary

Further information on research design is available in the Nature Portfolio Reporting Summary linked to this article.

## Data availability

The data supporting the findings of this study are available within the article and in Supplementary Data 1. Any other relevant information is available from the corresponding author upon reasonable request.

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

## Acknowledgements
The authors thank Benedikt Sabaß and Antonella Di Concilio Moschen for discussing theoretical considerations of α-catenin turnover in cells. This work was supported by the German Research Foundation's Priority Program SPP1782 and the Collaborative Research Consortium SFB 1348 (Project A12).

## Author contributions
L.W. performed all experiments and analyzed the data. C.G. supervised the study and wrote the paper with input from L.W.

## Funding

## Competing interests
The authors declare no competing interests.
