## [Transparent Peer Review file · Communications Biology]

A conformational change in α -catenin's actin-binding domain governs adherens junction maturation

Corresponding Author: Professor Carsten Grashoff

Version 0:

Reviewer comments:

Reviewer #1

(Remarks to the Author)

This work addressed yet not fully understand molecular complex reorganization during adherens junction maturation/turnover, which depends both on biochemical reaction (classical signalling pathways) and mechanics (mechanotransduction/mechanosensing). Given the impact of the adherens junction formation and remodelling during development, in adult tissues and in diseases the question remains on of the major one in the whole life science field. At the center of adherens junction mechanosensing is alpha-catenin, a mandatory partner of cadherin which unfolds under force to allow in particular vinculin binding and which interaction with F-actin is also force-dependent governed by a catch bond mechanism, although the two conformational changes underlying these forces dependence have not been demonstrated to work synergistically. A FRET sensor probe has been previously used to infer alpha-catenin force-dependent change in conformation in the vinculin binding modulation domain. However, it is still an issue to strictly demonstrate whether the types of probes are relating changes in conformation or force sensors.

Here the authors described a similar probe to the one developed previously by basically incorporating the sensor in another part of alpha-catenin, in Nter of the actin binding domain, close to H0 and H1 helices known to modulate the F-actin binding domain of alpha-catenin and proposed to contribute to allosteric stabilization under force of the cadherin-catenin complex to actin. Their first FRET measurement indeed revealed a conformational change in alpha-catenin in keratinocytes with non-mature and mature cell-cell junctions, but which was not correlated with forces.

The authors who are expert in this kind of probe (the main author developed the well-known vinculin based-FRET force sensor) further improved the method of analysis by applying FRET, FLIM and fluorescence anisotropy measurement and using force insensitive derivate of their probe. Using these tools combined with other mutations in the vinculin binding domain, they are able show that alpha-catenin undergoes a conformational change in this region close to the actin-binding domain, which is insensitive to vinculin binding, but requires actin polymerization and affects alpha-catenin turnover. The experiments are well done and illustrated. Results are convincing and support major conclusions, although the present data do not allow to make a comprehensive link between intramolecular tension and this conformational change.

Major points:

It would be nice if authors were verifying in at least another cell line.

What would happen if cadherin homophilic interaction are not engaged? For example: in undifferentiated cells (without Ca²⁺), or at cell contact free membrane. Indeed, it was reported using a so-called FRET based tension sensor on E-cadherin, that the complex looked as under tension even in cell membranes free of contact.

I doubt the last sentence of the result section (line 214-218) is supported by data.

Minor points:

All the mutants used in the study should be presented in a single figure.

Line 110-112, the ref to sup Fig 2 does not apply to the whole sentence, only to the second part. It illustrates silencing of alpha-cat only.

Not sure that mutants presented lines 114-116 are presented on a figure!

Reviewer #2

(Remarks to the Author)

In this manuscript by Windgasse and Grashoff, a tension sensor to detect conformational change in a-catenin was

developed by insertion of the FRET pair into the α -catenin C-terminal actin binding domain (ABD). Conformational changes during AJ maturation in epithelial sheets was investigated using a FLIM-FRET-based measurements. The key finding here is that α -catenin undergoes a conformational change in its ABD during the transition from punctate to linear form of AJs. The authors also showed that such conformational change depends on the organization of the actin network but not on the binding of vinculin. Furthermore, they observed that the conformational change in ABD correlates surprisingly with an increased turnover of α -catenin. This work addressed a central molecular link in cell-cell interactions and raise interesting and surprising points while the tool introduced should be generally useful to the field. However, the manuscript can be strengthened further by a number of key control experiments. Additionally, further quantitative comparison may be desirable as in its current form, there are aspects of this manuscript that seems highly descriptive.

Major points:

1. The study solely relies on the results obtained in murine epidermal keratinocytes, which can show different behavior from simple epithelia. The authors may want to demonstrate in more than one epithelial cell line in parallel to ensure that the observations are not dependent on the specific cell line and to evaluate how much the finding can be applied to epithelia in general.
2. Related to #1, a number of FRET sensors based on α -catenin have been developed previously by Kim et al., and Acharya et al., as mentioned by the authors. These studies applied their sensors in different cell types, such as DLD1 and MDCK (Kim et al.) or Caco2 (Acharya et al.). Given the surprising nature of the findings in the current study, e.g. the stability of α -catenin & the roles of vinculin, it would be helpful to perform comparisons whereby the FRET sensors developed here are used in cell lines used in previous studies, and conversely testing the Kim et al., and Acharya et al. FRET sensors in the MEK cells used in this study.
3. Figure 1g: The description of the results showing a wide range of FRET efficiency in both FL-TS-697 and F7-NF-697 (which is expected to be non-force-sensitive) seems questionable and would benefit from additional corroboration. To what extent is the variation in FL-TS-697 dependent on force? Is the F7-NF-697 mutant actually non-force-sensitive? I would suggest performing force calibration using drugs like CalyculinA and blebbistatin, as Acharya et al. did in the work in Dev Cell 2018. DOI: 10.1016/j.celrep.2017.02.078
4. Related to the comment above (#3), the authors exclude the contribution of mechanical forces from their considerations for further investigations. However, α -catenin is widely recognized as a mechanotransducer that undergoes conformational change induced by tensile force, and it is not likely reasonable to neglect its contribution. It should be more carefully justified experimentally, for example, by comparing with other α -catenin FRET sensors developed previously (#2)
5. Figure 2: Although the transition from PAs into MAs is a highly dynamic process, they only present the initial and terminal points. How are the puncta integrated into single linear junctions, and α -catenin molecules change their conformation spatiotemporally? Even given the limitations of FLIM imaging on temporal resolution, at least they can show some snapshots during the transition.
6. Figure 3: I do not clearly see PAs in the representative images (c) like shown in (a). Vehicle and washout controls are missing. Since cytochalasin D globally inhibits actin polymerization and thus cannot exclude the side effects, the authors can also consider performing knockdown/knockout of p114RhoGEF, which is responsible for circumferential actin belt formation. DOI: 10.1038/ncb2156
7. Figure 4: Nishimura et al. reported that the tension-sensitive vinculin binding states do not alter ZA/TJ formation but do PA morphology/number and speed of PA/ZA transition. Similar to the comment in Figure 2 (#4), the differential involvement of vinculin and F-actin binding during the transition process should be discussed in more depth. DOI: 10.1247/csf.22014
8. The increased turnover of α -catenin seems to be counterintuitive when considered against the prevailing model of cadherin-b-catenin- α -catenin-vinculin-actin. However, recent study (Morales et al. Nat Comm. 2024) suggests that the molecular mechanisms underlying cadherin-catenin mechanotransduction may be more complexed than initially thought. It may be worth discussing whether the findings here could make sense in such alternative mechanism.
9. It would help if the mechanisms proposed here can be summarized in a schematic drawing.

Minor points:

9. Figure 5: In the Discussion on Page 11, Lines 264 and 265, the authors mentioned "as bleached areas show uniform fluorescence recovery." I think this is an interesting and important observation, but it is not adequately presented as data in Figure 5a and b. I would suggest adding a kymograph of the bleached area.
10. Page 26, Line 657: L334P should be L344P.

Reviewer #3

(Remarks to the Author)

The article by Windgasse and Grashoff presents the observation of a conformational change in the ABD of alpha-catenin in living cells. The authors developed a FRET sensor to detect these conformational changes and demonstrated that they occur during adherens junction maturation. Overall, the article is well-written and scientifically sound. It is well established that the MD of alpha-catenin undergoes a force-induced conformational change that facilitates vinculin binding. The discovery of a second conformational change in the ABD of alpha-catenin is valuable for understanding the function and force regulation of adherens junctions. I believe this study will be of interest to the readers of Communications Biology. However, there are several points that need to be addressed:

1. In the introduction (lines 39–46), the authors discuss the hypothesis that a force-induced conformational change in two alpha-helices within the C-terminal ABD may be responsible for the catch bond with actin, citing a series of articles in support of this hypothesis (refs 3, 9–13). While these references are relevant, the current wording suggests that all cited studies directly support this hypothesis, which is not the case. This section should be revised to more clearly distinguish the specific contributions of each reference. For instance, to my knowledge, the first study proposing a conformational change in the ABD is Xu et al., eLife 2020 (which is not cited). Ref 9 demonstrates a catch bond between alpha-catenin and actin. Ref

13 shows that the catch bond is cooperative, occurring only in the presence of multiple molecules. Ref 10 provides evidence that the ABD of alpha-catenin forms a catch bond with actin, while refs 13 (and later ref 11) establishes that this catch bond is directionally asymmetric.

2. The authors state that their data “show that the actin-bound, presumably stabilized α -catenin catch-bond conformation correlates with an overall increased rather than decreased protein turnover at cell-cell junctions” (lines 56–57). This conclusion is based on FRAP experiments described on pages 8–9, which compare the mobile fractions between PA and MA. However, the term protein turnover typically refers to the rate at which old proteins are replaced by newly synthesized ones. Here, it seems the authors intend to describe the rate of alpha-catenin exchange at adherens junctions. Additionally, FRAP recovery curves often do not follow a simple exponential function, and the mobile fraction at a single time point (560s) does not directly indicate the exchange rate. Ideally, FRAP experiments should be conducted over a longer duration to capture the fluorescence signal plateau, and the data should be fitted with an appropriate kinetic model to extract meaningful exchange rate information (see Sprague and McNally, Trends in Cell Biology, 2005). For example, ref 6 followed a similar approach and found that the FRAP half-time remained relatively constant.

3. In Fig. 4d, the authors compare the MD sensors between MA and cytochalasin-treated cells. However, to confirm that the MD conformational change is not involved, they should directly compare PA and MA cells.

4. The authors should provide a clearer explanation of the FAIM data to aid readers who are less familiar with the technique.

Version 1:

Reviewer comments:

Reviewer #1

(Remarks to the Author)

The authors positively answered my comments

Reviewer #2

(Remarks to the Author)

The authors have addressed my comments satisfactorily. I now support its publication in Communications Biology.

Reviewer #1 (Remarks to the Author):

This work addressed yet not fully understand molecular complex reorganization during adherens junction maturation/turnover, which depends both on biochemical reaction (classical signalling pathways) and mechanics (mechanotransduction/mechanosensing). Given the impact of the adherens junction formation and remodelling during development, in adult tissues and in diseases the question remains on of the major one in the whole life science field. At the center of adherens junction mechanosensing is alpha-catenin, a mandatory partner of cadherin which unfolds under force to allow in particular vinculin binding and which interaction with F-actin is also force-dependent governed by a catch bond mechanism, although the two conformational changes underlying these forces dependence have not been demonstrated to work synergistically. A FRET sensor probe has been previously used to infer alpha-catenin force-dependent change in conformation in the vinculin binding modulation domain. However, it is still an issue to strictly demonstrate whether the types of probes are relating changes in conformation or force sensors.

Here the authors described a similar probe to the one developed previously by basically incorporating the sensor in another part of alpha-catenin, in Nter of the actin binding domain, close to H0 and H1 helices known to modulate the F-actin binding domain of alpha-catenin and proposed to contribute to allosteric stabilization under force of the cadherin-catenin complex to actin. Their first FRET measurement indeed revealed a conformational change in alpha-catenin in keratinocytes with non-mature and mature cell-cell junctions, but which was not correlated with forces.

The authors who are expert in this kind of probe (the main author developed the well-known vinculin based-FRET force sensor) further improved the method of analysis by applying FRET, FLIM and fluorescence anisotropy measurement and using force insensitive derivate of their probe. Using these tools combined with other mutations in the vinculin binding domain, they are able show that alpha-catenin undergoes a conformational change in this region close to the actin-binding domain, which is insensitive to vinculin binding, but requires actin polymerization and affects alpha-catenin turnover.

The experiments are well done and illustrated. Results are convincing and support major conclusions, although the present data do not allow to make a comprehensive link between intramolecular tension and this conformational change.

Response: We thank the reviewer for taking the time to carefully evaluate our manuscript and for the very helpful and encouraging remarks. We have addressed all the points raised by the reviewer, by performing additional experiments, generating new figures, and adjusting the text, as requested. Together with the additional changes requested by the other reviewers, we think that this has further improved the manuscript and we hope that the reviewer can fully support the publication of our study.

It would be nice if authors were verifying in at least another cell line.

Response: We think that this is a very good suggestion, which has been also brought up by the second reviewer, who suggested experiments in MDCK cells. We therefore repeated our experiments in this cell type, and the resulting data confirm our findings from keratinocytes. We also observe a wide spread of

FRET efficiencies in cell-cell junctions of MDCK cells when the conformation sensitive probe is used, while control constructs with a C-terminal fusion of the same FRET module display the normal, narrow spread in FRET efficiencies. These additional data indicate that the here observed defect occurs in simple and stratifying epithelia. The new data are now shown in the updated Fig. 2 (k-m) and described in the text in lines 146-153.

What would happen if cadherin homophilic interaction are not engaged? For example: in undifferentiated cells (without Ca²⁺), or at cell contact free membrane. Indeed, it was reported using a so-called FRET based tension sensor on E-cadherin, that the complex looked as under tension even in cell membranes free of contact.

Response: In response to this suggestion, we performed experiments in which we focussed on α -catenin signals at the contact-free membrane. However, in contrast to E-cadherin, which can be detected, albeit at low intensity, at the free membrane, α -catenin is not sufficiently enriched at the membrane to perform reasonable FRET measurements (see Fig.1 for reviewer 1). We would prefer not to show the data in the manuscript as these datasets do not provide any additional information. We hope that the reviewer will agree.

I doubt the last sentence of the result section (line 214-218) is supported by data.

Response: We deleted this sentence to avoid any misunderstanding.

Minor points:

All the mutants used in the study should be presented in a single figure.

Response: Following the suggestion of the reviewer, we prepared a new figure in which all expression constructs are schematically depicted and described in the figure legends. This figure is now shown as Supplementary Fig. 2 and we refer to it throughout the manuscript.

Line 110-112, the ref to sup Fig 2 does not apply to the whole sentence, only to the second part. It illustrates silencing of alpha-cat only.

Response: We have moved the reference to the end of the sentence (line 118). Please note that the data showing the α -catenin knockdown are now shown in Supplementary Fig. 4.

Not sure that mutants presented lines 114-116 are presented on a figure!

Response: All constructs are now depicted in the additional Supplementary Fig. 2.

Reviewer #2 (Remarks to the Author):

In this manuscript by Windgasse and Grashoff, a tension sensor to detect conformational change in α -catenin was developed by insertion of the FRET pair into the α -catenin C-terminal actin binding domain (ABD). Conformational changes during AJ maturation in epithelial sheets was investigated using a FLIM-FRET-based measurements. The key finding here is that α -catenin undergoes a conformational change in its ABD during the transition from punctate to linear form of AJs. The authors also showed that such conformational change depends on the organization of the actin network but not on the binding of vinculin. Furthermore, they observed that the conformational change in ABD correlates surprisingly with an increased turnover of α -catenin. This work addressed a central molecular link in cell-cell interactions and raise interesting and surprising points while the tool introduced should be generally useful to the field. However, the manuscript can be strengthened further by a number of key control experiments. Additionally, further quantitative comparison may be desirable as in its current form, there are aspects of this manuscript that seems highly descriptive.

Response: We thank the reviewer for carefully evaluating our manuscript and the helpful and very constructive suggestions. We have performed additional experiments and assembled new sets of data, adjusted the main text and figures. We think that these changes, together with a number of additional experiments requested by the other reviewers, further improved the manuscript. We hope the reviewer can now support the publication of these data sets.

1. The study solely relies on the results obtained in murine epidermal keratinocytes, which can show different behavior from simple epithelia. The authors may want to demonstrate in more than one epithelial cell line in parallel to ensure that the observations are not dependent on the specific cell line and to evaluate how much the finding can be applied to epithelia in general.

Response: Analysing an additional cell line has been also suggested by another reviewer and we chose to use MDCK cells, as a model for simple epithelia. The obtained results are consistent with our previous observations in keratinocytes. Again, we observe a comparably wide spread of FRET efficiencies in cell-cell junctions of MDCK cells when the conformation sensor is used, while control constructs with a C-terminal fusion of the same FRET module display the normal spread in FRET efficiencies. These data indicate that the observed conformational change occurs in simple and stratifying epithelia and is, indeed, a more general phenomenon. The new data are now shown in the updated Fig. 2 (k-m) and described in the text in lines 146-153.

2. Related to #1, a number of FRET sensors based on α -catenin have been developed previously by Kim et al., and Acharya et al., as mentioned by the authors. These studies applied their sensors in different cell types, such as DLD1 and MDCK (Kim et al.) or Caco2 (Acharya et al.). Given the surprising nature of the findings in the current study, e.g. the stability of α -catenin & the roles of vinculin, it would be helpful to perform comparisons whereby the FRET sensors developed here are used in cell lines used in previous

studies, and conversely testing the Kim et al., and Acharya et al. FRET sensors in the MEK cells used in this study.

Response: Following the suggestion of the reviewer, we performed experiments using the here developed FRET sensor in MDCK cells, and analysed the constructs described in the Acharya et al study in MEKs. Please note that the sensor described in Kim et al. was already used in MEKs, shown in Fig. 4.

Using the Acharya et al. constructs in MEKs yielded results that are consistent with our previous findings. While the C-terminally truncated α -Cat TL construct from the Archaya study showed a comparatively small variance in FRET efficiency, we observed a significantly larger spread in FRET efficiency using the α -Cat TS probe (which is based on an F40-linker-based mTFP Venus sensor). Although the differences are not as pronounced as for our Ypet-F7-mCherry-based constructs, we also observe higher FRET efficiencies in punctate adhesions and significantly lower efficiencies in mature adhesions of differentiating keratinocytes. The results of these experiments are shown below.

While these data are consistent with our findings, the results contradict the previous work of Archaya et al. who observed higher FRET efficiencies with the α -Cat TL construct and lower values with the α -Cat TS probe. We observed the exact opposite effect. Although we are confident in our experimental procedure, performed the experiments with a sufficiently high number of cells (recorded on three independent experimental days), and the data are statistically sound, we do not wish to emphasize these findings in the manuscript. Testing whether the Archaya et al. constructs can be used as tension sensors in MEKs would require months of additional experiments, but the answer to this particular question is not the focus of our study.

3. Figure 1g: The description of the results showing a wide range of FRET efficiency in both FL-TS-697 and F7-NF-697 (which is expected to be non-force-sensitive) seems questionable and would benefit from additional corroboration. To what extent is the variation in FL-TS-697 dependent on force? Is the F7-NF-697 mutant actually non-force-sensitive? I would suggest performing force calibration using drugs like

CalyculinA and blebbistatin, as Acharya et al. did in the work in Dev Cell 2018. DOI: 10.1016/j.celrep.2017.02.078

Response: We thank the reviewer for this comment because it revealed that we did not explain our rationale properly. We do not wish to argue that mechanical forces are not important for the observed effects. What we would like to emphasize in our study is that the F7-NF-697 FRET module is force-insensitive (i.e., the F7 peptide is too short to mediate a force-sensitive response of the FRET probe) and therefore suited to monitor conformation-specific effects. To make this clearer we slightly adjusted the text (lines 107-1110).

As suggested by the reviewer, we performed the blebbistatin and calyculin-A experiments in a culture of cells showing punctate and mature adhesions, to ensure that experimental conditions were identical for PA and MA. Consistent with our hypothesis, we still observed higher FRET efficiencies in PA and low FRET values in MA in cells expressing F7-NF-697 conformation sensor, while the C-terminal control remained unaltered at around 20 % FRET efficiency. These data support the assumption that the conformation sensor does not directly report on changes in actomyosin tension. The new data are now included in a new Supplementary Fig. 3 and described in lines 112-115.

4. Related to the comment above (#3), the authors exclude the contribution of mechanical forces from their considerations for further investigations. However, α -catenin is widely recognized as a mechanotransducer that undergoes conformational change induced by tensile force, and it is not likely reasonable to neglect its contribution. It should be more carefully justified experimentally, for example, by comparing with other α -catenin FRET sensors developed previously (#2)

Response: This is related to the point above. We do not want to argue that mechanical forces do not play a role in α -catenin regulation, nor do we want to neglect the contribution of these forces. What we want to communicate is that α -catenin undergoes a conformational change at the ABD and that this can be specifically monitored with the F7-NF-697 construct. Please note that we have designed and performed the rather complex anisotropy experiments to confirm this assumption. In these anisotropy experiments, we use a donor-only construct that cannot undergo FRET changes in response to force. Nevertheless, we observe differences. This is direct evidence that the probe itself does not respond to mechanical forces, but reflects a conformational change. Again, this does not mean that α -catenin or the overall process itself is force-insensitive, and nowhere in the manuscript do we make that statement.

5. Figure 2: Although the transition from PAs into MAs is a highly dynamic process, they only present the initial and terminal points. How are the puncta integrated into single linear junctions, and α -catenin molecules change their conformation spatiotemporally? Even given the limitations of FLIM imaging on temporal resolution, at least they can show some snapshots during the transition.

Response: As requested, we acquired snapshot images of cells at intermediate stages. These data suggest that the fraction of α -catenin molecules being in the mature conformation progressively increases during differentiation. These data are now shown in the new Fig. 2 for MEKs and for MDCK cells.

6. Figure 3: I do not clearly see PAs in the representative images (c) like shown in (a). Vehicle and washout controls are missing. Since cytochalasin D globally inhibits actin polymerization and thus cannot exclude the side effects, the authors can also consider performing knockdown/knockout of p114RhoGEF, which is responsible for circumferential actin belt formation. DOI: 10.1038/ncb2156

Response: We now show higher magnification images in the new Figure 3, which better illustrate that starvation and cytochalasin-D treatment induce a PA-like phenotype, characterized by a punctate appearance of the E-cadherin signal. Please note the difference to Taxol-treated cells, which show a very distinct E-cadherin signal at cell-cell junctions. We also performed washout experiments as requested and these data are now shown in the new Supplementary Figure 6. These experiments confirm that the concentrations of DMSO used in these experiments do not cause significant changes in the overall appearance of cells and AJs. In addition, the data show that the effects of Cytochalasin-D are reversible and that actin organization is restored after 1 h. Taxol washout does not induce a significant effect, presumably because AJs have been induced to a mature state and are associated with a cortical actin structure, a state in which they remain after removal of the compound.

7. Figure 4: Nishimura et al. reported that the tension-sensitive vinculin binding states do not alter ZA/TJ formation but do PA morphology/number and speed of PA/ZA transition. Similar to the comment in Figure 2 (#4), the differential involvement of vinculin and F-actin binding during the transition process should be discussed in more depth. DOI: 10.1247/csf.22014

Response: We have tried to incorporate the findings by Nishimura et al in a reasonable way, but we are somewhat hesitant to draw conclusions from comparing our studies, which use different cells and cell culture systems. Not considering potential cell type specific effects, it seems to us that the previously described conformation change in the MD of α -catenin, which seem to be tension-sensitive and facilitates vinculin recruitment, and the here described structural rearrangement at the ABD both promote and/or correlate with junction maturation, but can occur independently.

8. The increased turnover of α -catenin seems to be counterintuitive when considered against the prevailing model of cadherin-b-catenin-a-catenin-vinculin-actin. However, recent study (Morales et al. Nat Comm. 2024) suggests that the molecular mechanisms underlying cadherin-catenin mechanotransduction may be more complexed than initially thought. It may be worth discussing whether the findings here could make sense in such alternative mechanism.

Response: We thank the reviewer for this insightful remark. Indeed, it seems important to note that other force bearing linkages exist, such as the E-Cadherin- β -catenin-vinculin linkage proposed by Morales et al. We included this aspect in the discussion as suggested, and also refer to the Morales study in figure legends of the new Fig. 6.

9. It would help if the mechanisms proposed here can be summarized in a schematic drawing.

Response: Following the advice of the reviewer, we now include a schematic drawing as Figure 6 summarizing the main findings. The picture illustrates that α -catenin can adopt distinct conformations, depending on the association with the actin network, which can be monitored with the F7-NF-697 probe. The figure also shows that these distinct structures are characterized by different turnover rates.

Minor points:

9. Figure 5: In the Discussion on Page 11, Lines 264 and 265, the authors mentioned “as bleached areas show uniform fluorescence recovery.” I think this is an interesting and important observation, but it is not adequately presented as data in Figure 5a and b. I would suggest adding a kymograph of the bleached area.

Response: We have modified Figure 5 to include a kymograph analysis of the fluorescence recovery. The data provide visual confirmation that the recovery is uniform.

10. Page 26, Line 657: L334P should be L344P.

Response: The typo was corrected.

Reviewer #3 (Remarks to the Author):

The article by Windgasse and Grashoff presents the observation of a conformational change in the ABD of alpha-catenin in living cells. The authors developed a FRET sensor to detect these conformational changes and demonstrated that they occur during adherens junction maturation. Overall, the article is well-written and scientifically sound. It is well established that the MD of alpha-catenin undergoes a force-induced conformational change that facilitates vinculin binding. The discovery of a second conformational change in the ABD of alpha-catenin is valuable for understanding the function and force regulation of adherens junctions. I believe this study will be of interest to the readers of Communications Biology.

However, there are several points that need to be addressed:

Response: We would like to thank the reviewer for taking the time to carefully evaluate our manuscript and making very constructive and insightful suggestions. In response to these comments, we adjusted the text and also performed additional experiments. We think that these changes, together with a number of additional experiments requested by the other reviewers, further improved the manuscript. We hope the reviewer can now support the publication of these interesting data.

1. In the introduction (lines 39–46), the authors discuss the hypothesis that a force-induced conformational change in two alpha-helices within the C-terminal ABD may be responsible for the catch bond with actin, citing a series of articles in support of this hypothesis (refs 3, 9–13). While these references are relevant, the current wording suggests that all cited studies directly support this hypothesis, which is not the case. This section should be revised to more clearly distinguish the specific contributions of each reference. For instance, to my knowledge, the first study proposing a conformational change in the ABD is Xu et al., eLife 2020 (which is not cited). Ref 9 demonstrates a catch bond between alpha-catenin and actin. Ref 13 shows that the catch bond is cooperative, occurring only in the presence of multiple molecules. Ref 10 provides evidence that the ABD of alpha-catenin forms a catch bond with actin, while refs 13 (and later ref 11) establishes that this catch bond is directionally asymmetric.

Response: We thank the reviewer for this suggestion and agree that the original work should have been cited more precisely. We adjusted the introduction accordingly.

2. The authors state that their data “show that the actin-bound, presumably stabilized α -catenin catch-bond conformation correlates with an overall increased rather than decreased protein turnover at cell-cell junctions” (lines 56–57). This conclusion is based on FRAP experiments described on pages 8–9, which compare the mobile fractions between PA and MA. However, the term protein turnover typically refers to the rate at which old proteins are replaced by newly synthesized ones. Here, it seems the authors intend to describe the rate of alpha-catenin exchange at adherens junctions. Additionally, FRAP recovery curves often do not follow a simple exponential function, and the mobile fraction at a single time point (560s)

does not directly indicate the exchange rate. Ideally, FRAP experiments should be conducted over a longer duration to capture the fluorescence signal plateau, and the data should be fitted with an appropriate kinetic model to extract meaningful exchange rate information (see Sprague and McNally, Trends in Cell Biology, 2005). For example, ref 6 followed a similar approach and found that the FRAP half-time remained relatively constant.

Response: We thank the reviewer for this thoughtful comment. We agree that the FRAP experiments should have been conducted over a longer duration to reach the plateau phase. By combining FLIM-FRET with FRAP experiments in one experiment, we had been somewhat limited ourselves because photobleaching became an issue. We therefore complemented the existing data sets with a new FRAP experiment that extends the time of observation to 20 min. Even more extended durations were not possible because the punctate adhesions start to move or disassemble and therefore cannot be reliably tracked for longer times. However, these new data, now shown in Fig. 5f, capture the plateau phase much better and allow bi-exponential fitting with high quality. The data are not only consistent with the previous FRAP-FLIM data sets, showing higher turnover rates of MA, but they also indicate that at least two distinct components fuel the α -catenin recovery.

3. In Fig. 4d, the authors compare the MD sensors between MA and cytochalasin-treated cells. However, to confirm that the MD conformational change is not involved, they should directly compare PA and MA cells.

Response: This is a good suggestion, and of course we initially tried to perform the experiments as suggested. Unfortunately, the MD sensor published by Kim et al. does not efficiently localize to PA in MEKs, indicating that insertion of the fluorophores does compromise α -catenin function to some extent. Please note that we just reproduced the previously reported construct (Kim et al., 2015), which was originally tested in simple epithelial cells (MDCK) but not in stratifying epithelial cells. As the MD sensor is efficiently localizing to MA, we decided to induce PA-like adhesions through starvation and CytoD treatment (as shown in Fig. 3).

4. The authors should provide a clearer explanation of the FAIM data to aid readers who are less familiar with the technique.

Response: We now better explain the rationale behind FAIM measurements and also refer to an insightful paper (Devauges et al, 2012) that provides more detailed information on FAIM measurements.